# From Cell to Symptoms: The Role of SARS-CoV-2 Cytopathic Effects in the Pathogenesis of COVID-19 and Long COVID

**DOI:** 10.3390/ijms24098290

**Published:** 2023-05-05

**Authors:** Pablo Gonzalez-Garcia, Ornella Fiorillo Moreno, Eloina Zarate Peñata, Alejandro Calderon-Villalba, Lisandro Pacheco Lugo, Antonio Acosta Hoyos, Jose Luis Villarreal Camacho, Roberto Navarro Quiroz, Leonardo Pacheco Londoño, Gustavo Aroca Martinez, Noelia Moares, Antonio Gabucio, Cecilia Fernandez-Ponce, Francisco Garcia-Cozar, Elkin Navarro Quiroz

**Affiliations:** 1Institute of Biomedical Research Cadiz (INIBICA), 11009 Cádiz, Spain; pablo.gonzalezgarcia@alum.uca.es (P.G.-G.); ceciliamatilde.fernandez@uca.es (C.F.-P.); curro.garcia@uca.es (F.G.-C.); 2Clínica Iberoamerica, Barranquilla 080001, Colombia; ornella.fiorillo11@gmail.com; 3Life Science Research Center, Universidad Simon Bolívar, Barranquilla 080001, Colombia; eloina.zarate@unisimon.edu.co (E.Z.P.); lisandro.pacheco@unisimon.edu.co (L.P.L.); antonio.acosta@unisimon.edu.co (A.A.H.); leonardo.pacheco@unisimon.edu.co (L.P.L.); gustavo.aroca@unisimon.edu.co (G.A.M.); 4Department of Nutrition and Bromatology, Pablo de Olavide University, 41013 Seville, Spain; acalvil@alu.upo.es; 5School of Medicine, Universidad Libre, Barranquilla 080001, Colombia; josel.villarrealc@unilibre.edu.co; 6Department of Structural and Molecular Biology, Molecular Biology Institute of Barcelona, Spanish National Research Council, 08028 Barcelona, Spain; robertcnavarro@gmail.com; 7School of Medicine, Universidad del Norte, Barranquilla 080001, Colombia; 8Department of Biomedicine, Biotechnology and Public Health, Faculty of Medicine, University of Cadiz, 11003 Cádiz, Spain; noelia.moares@gm.uca.es (N.M.); antonio.gabucio@uca.es (A.G.)

**Keywords:** COVID-19, SARS-CoV-2, long COVID, cytopathy, cytokine storm, sequelae, PASC, coronavirus, angiotensin-converting enzyme 2, cell dysfunction

## Abstract

Severe Acute Respiratory Syndrome CoronaVirus 2 (SARS-CoV-2) infection triggers various events from molecular to tissue level, which in turn is given by the intrinsic characteristics of each patient. Given the molecular diversity characteristic of each cellular phenotype, the possible cytopathic, tissue and clinical effects are difficult to predict, which determines the heterogeneity of COVID-19 symptoms. The purpose of this article is to provide a comprehensive review of the cytopathic effects of SARS-CoV-2 on various cell types, focusing on the development of COVID-19, which in turn may lead, in some patients, to a persistence of symptoms after recovery from the disease, a condition known as long COVID. We describe the molecular mechanisms underlying virus-host interactions, including alterations in protein expression, intracellular signaling pathways, and immune responses. In particular, the article highlights the potential impact of these cytopathies on cellular function and clinical outcomes, such as immune dysregulation, neuropsychiatric disorders, and organ damage. The article concludes by discussing future directions for research and implications for the management and treatment of COVID-19 and long COVID.

## 1. Introduction

The infectious agent of Severe Acute Respiratory Syndrome Coronavirus 2 (SARS-CoV-2) that causes coronavirus disease 2019 (COVID-19) is a single-stranded, positive-sense, enveloped RNA virus of the genus Betacoronavirus [1]. The SARS-CoV-2 genome (GenBank accession NC_045512) has 29,903 bases encoding 14 open reading frames (ORFs), which code for 27 different proteins [2]. Two large open reading frames (ORFs), ORF1a and ORF1b, constitute two-thirds of the genome and are translated into pp1a and pp1ab, which are two large polypeptides that encompass all non-structural proteins (nsps) of the virus and play diverse roles in viral replication [1]. Two of them, nsp3 and nsp5, are viral cysteine proteases that cleave pp1a and pp1ab polypeptides into 16 nonstructural proteins (nsp1–nsp16). The other third of the genome has overlapping ORFs, encoding 4 structural proteins, S (spike glycoprotein), N (nucleocapsid protein), M (membrane protein), and E (envelope protein), as well as 8 other accessory proteins (ORF3a, ORF6, ORF7a, ORF7b, ORF8, ORF9b, ORF9c, and ORF10) [2,3,4].

SARS-CoV-related coronaviruses initiate their entry into the host cell by binding protein S on the virus to specific cell membrane proteins. The tropism of the different coronaviruses is given by the interaction of the S-protein homotrimer with host cell surface molecules, such as CLEC4M/DC-SIGNR (C-type lectin domain family 4 member or CD299) for SARS-CoV [5] hACE2 (human angiotensin-converting enzyme 2), Aminopeptidase N (APN) [6,7] for SARS-CoV and SARS-CoV-2, Dipeptidyl peptidase 4 (DPP4) [7] for MERS-CoV, as well as Neuropilin-1 (NRP1) [8,9] and carcinoembryonic antigen-related cell adhesion molecule 1 (CEACAM1) [10] for SARS-CoV-2, although the main target for the SARS-CoV-2 spike protein is the ACE2 receptor, as well as the serine protease TMPRSS2, which is used for spike protein priming [11,12]. This interaction determines the possibility of entry of the virus into the different cellular phenotypes of the host, as expressed by some of these molecules on their surface [13]. This interaction, which is similar among SARS-CoV-related coronaviruses, induces a conformational transition from prefusion to post-fusion that involves processing by proteases such as TMPRSS2 expressed in the cellular surface of the host cell [14] or cathepsins and furin in the endosomal compartment [15]. Then, a fusion pore is formed, enabling the fusion of both viral and cell membranes, thus allowing the viral genome to reach the cell cytoplasm [16]. Viral replication occurs initially in virus-induced double-membrane vesicles (DMV) derived from the endoplasmic reticulum (ER), which integrate to form elaborate networks of intricate membranes. The incoming viral genome serves as a template for nsp12 to synthesize viral genomic RNA, and all viral proteins are inserted into the ER-Golgi intermediate compartment (ERGIC) for assembly together with the virion genome [17,18]. The newly synthesized virions are then secreted across the plasma membrane, as occurs in other SARS-CoV-related viruses [19]. 

In addition, it has been widely reported that the consequences of SARS-CoV-2 infection go beyond replication within the respiratory system, leading to various extrapulmonary manifestations [20]. Therefore, virus replication in the human organism may have several consequences on cellular and/or tissue physiology, which vary depending on the cellular phenotype [21,22]. Defining the possible pathological mechanisms triggered by molecular interactions between virus and host in a tissue-specific manner could contribute to the understanding of the occurrence of the diversity of clinical outcomes of SARS-CoV-2 infection.

Although most COVID-19 survivors recover normally after viral clearance, there is a small percentage of patients who remain with sequelae for a variable length of time, a condition known as long COVID or Post-acute COVID-19 syndrome. This disease, which is estimated to affect about 10% of recovered non-hospitalized COVID-19 patients and 50–70% of hospitalized patients, has more than 200 symptoms and different levels of severity, making it a difficult disease to define [23]. Although multiple studies are ongoing to explain the molecular processes, it has been previously described that exacerbated cytokine production, a procoagulant state, and direct cellular damage caused directly or indirectly by SARS-CoV-2 infection may be involved in the development of long-term symptoms [24]. This article aims to provide an overview of how the SARS-CoV-2 virus can affect the structure and function of human cells during infection by establishing a broad spectrum of cytopathies, highlighting their direct and indirect effects on different cell types, as well as the interactions between viral proteins and host cells. In addition, we aim to explore how these cellular changes can contribute to the development of the long-term symptoms seen in long COVID patients.

## 2. Alterations in Cellular Structures and Organelles Due to SARS-CoV-2 Infection

Upon entry into the host cell, SARS-CoV-2 interacts with cellular molecules and modulates the metabolic activity of the cell, leading to various cytopathic effects. These effects are determined by a combination of factors, among which the host cell-specific proteome plays a key role. The virus affects cellular organelles in ways that are similar to other coronaviruses. A better understanding of these interactions may help to better understand how SARS-CoV-2 causes host cell damage and contributes to the development of various COVID-19- and long COVID-associated symptoms. In the following section, we will describe some of these interactions in more detail, highlighting their impact on different cellular structures.

### 2.1. Cytopathic Effects on Mitochondria

Mitochondria form networks within the cell and can change their shape depending on the cellular energy needs [25]. It has been shown that the mitochondrial network is susceptible to alterations in its dynamics and function through the interaction between SARS-CoV-2 proteins and certain mitochondrial proteins. Mitochondria could be one of the organelles most affected by SARS-CoV-2-derived cytopathy, as their alteration can lead to cellular stress, which in turn has several consequences depending on the cellular phenotype. Different interactions between viral and mitochondrial molecules generate alterations in the mitochondrial antiviral response, the induction of different phenotype-specific programmed cell death programs, the catabolism of amino acids and lipids, and the level of energy obtained through aerobic cellular respiration so that they lose functionality and accumulate damage. All this can be considered a collateral effect of cellular sequestration by the virus to redirect cellular metabolism toward viral replication. Although this condition usually reverses after viral clearance, its chronification may be associated with symptoms of mitochondrial-associated long COVID, which in turn can be a consequence of previous suboptimal mitochondrial function [26]. In this sense, the cellular mechanisms that eliminate damaged mitochondria lose their effectiveness and, consequently, they accumulate in the cells. Previous studies have reported that the accumulation of damaged mitochondria and oxidative stress are common factors in the development of various clinical manifestations of long COVID, such as chronic fatigue syndrome, “brain fog”, and cognitive impairment [27,28,29]. In fact, the accumulation of damaged mitochondria in pulmonary artery smooth muscle cells and airway epithelial cells has been hypothesized as a consequence of direct hypoxemia and alveolar injury by SARS-CoV-2 infection, thus inducing hypoxic pulmonary vasoconstriction, which is altered in some COVID-19 patients [30].

One of these cytopathic effects is that SARS-CoV-2 can also alter both the synthesis and transport of mitochondrial proteins. On the one hand, the nsp8 protein, which has been described to interact with neuroguidine (NGDN), asparaginyl-tRNA synthetase 2 (NARS2), and mitochondrial ribosomal protein S5, 25, and 27 (MRPS5, 25, and 27), could alter the modulation of mitochondrial protein synthesis [31,32,33], as well as interact with the ETC complex [34]. On the other hand, the nsp10 protein could modulate protein transport from the inner mitochondrial membrane to the mitochondrial matrix through its interaction with the host protein GRPEL1 (GrpE protein homolog 1) [35], whose functional significance remains unclear, although its loss has been described as leading to mitochondrial oxidation of fatty acids and arrest of oxidative phosphorylation in musculoskeletal cells, along with rapid muscle atrophy [36]. Moreover, the SARS-CoV-2 ORF9b protein alters responses to type I interferon (IFN-I) by interacting with Translocase of Outer Membrane 70 (TOM70), the consequences of which could benefit virus replication [37]. TOM70 participates in protein transport into mitochondria and establishes a communication pathway between mitochondria and the nucleus by connecting to the nuclear transcriptional activity of mitochondrial proteins [38]. When bound to ORF9b, TOM70 may not bind properly to HSP90, as TOM70-HSP90 binding is critical for the development of TOM70-mediated IFN-I activation, and TOM70 dysfunctions have been correlated with an induction of lactic acid production, the accumulation of which has been reported to inhibit IFN-I responses [39]. Interestingly, previous studies with SARS-CoV showed that its ORF9b protein causes morphological changes in mitochondria, such as their elongation, by promoting the degradation of dynamin-related protein 1 (DRP1), which is a GTPase responsible for the regulation of mitochondrial fission. These morphological alterations have been associated with autophagy.

Consistent with protein–host interactions whose consequences lie in mitochondrial dysfunctions, the SARS-CoV-2 nsp12 protein has been suggested to inhibit branched-chain keto acid dehydrogenase kinase (BCKDK), a branched-chain amino acid kinase regulatory enzyme [34,35]. Interestingly, the accumulation of branched-chain amino acids, which in turn can inhibit pyruvate dehydrogenase, mitochondrial respiration chain, and α-ketoglutarate dehydrogenase, potentially causes apoptosis in glial cells and neurons [40]. In addition, the nsp7 and nsp8 proteins have been described as two cofactors of the nsp12 protein, and both interact with ribosomal proteins in mitochondria [34]. 

In addition, other cytopathic effects can be deduced from the virus–host interactome. SARS-CoV-2 M protein may also alter other mitochondrial proteins, such as acyl-CoA medium-chain dehydrogenase (ACADM), coenzyme Q8B (COQ8B), pitrilysin metallopeptidase 1 (PITRM1), and the mitochondrial processing peptidase alpha and beta subunits (PMPCA and PMPCB), among others [35]. Although the M protein interacts with many mitochondrial proteins, other viral proteins can also alter the function of these organelles. The SARS-CoV-2 N protein interacts with eukaryotic translation initiation factor 4E (EIF4E2) family member 2 [35], which is a repressor of the translation initiation [27]. Interestingly, it has been described that both the M protein and the N protein of SARS-CoV are able to induce apoptosis in human lung fibroblast cells [28]. In addition, the SARS-CoV-2 nsp4 protein interacts with the inner mitochondrial membrane translocases 9, 10, and 29 (TIMM 9, 10, and 29), and the viral nsp6 protein interacts with the membrane ATP synthase subunit G (ATP5MG) [35], which is important in the synthesis of ATP in aerobic cellular respiration (Figure 1) [29]. Another non-structural protein of the virus, the SARS-CoV-2 nsp7 protein, when transduced into BEAS-2B cells, increases AFM1 (apoptosis-inducing factor 1, mitochondrial) gene expression, reduces both basal and maximal oxygen consumption rates, promotes mitochondrial fission, reduces electron transport chain (ETC) complex I activity, and inhibits oxidative metabolism [30], a phenomenon that occurs through the interaction of the nsp7 protein with some proteins with an important role in the electron transport process, such as NADH:Ubiquinone Oxidoreductase Complex Assembly Factors 1 (NDUFAF1) and 2 (NDUFAF2), NADH-cytochrome b5 reductase 3 (CYB5R3), and cytochrome b5 type B (CYB5B) [30,35]. Furthermore, Sirtuin 5 (SIRT5), which is a NAD-dependent protein deacylase critical for cellular metabolism [31], has been described as a putative SARS-CoV-2 nsp14 binding partner, which is an important interaction that allows for enhanced viral replication success [32,35]. Furthermore, the interaction of nsp14 with Sirtuin 1 (SIRT1) can inhibit its ability to activate the NRF2/HMOX1 pathway, thus dysregulating host antioxidant defense [33,41]. 

Thus, some of the aforementioned effects of SARS-CoV-2 proteins on mitochondria may lead to cellular stress, decreased energy production, and increased oxidative damage, which in turn could contribute to the development of some long COVID symptoms such as fatigue, “brain fog”, reduced exercise tolerance, and muscle weakness. 

### 2.2. Cytopathic Effects on the Endoplasmic Reticulum

The endoplasmic reticulum (ER), consisting of a network of interconnected channels and flattened sacs, plays a vital role in multiple cellular functions, such as protein synthesis, folding, and transport, and regulation of intracellular Ca^2+^ levels, among others. Known to play a role in the SARS-CoV-2 replicative cycle [17], the “hijacking” of this organelle by the viral replication machinery can trigger an ER stress response as a consequence of an altered accumulation of unfolded protein in the ER lumen, which in turn is due to exacerbated protein entry into the ER protein folding system. Persistent ER stress causes Ca^2+^ to outflow from the ER, which enters the mitochondria and triggers their release of cytochrome C, thereby activating caspases 3 and 9 and, in turn, triggering apoptosis. Therefore, persistent viral proliferation could be responsible for the induction of necroptosis, autophagy, caspase-dependent apoptosis, and activation of mitogen-activated protein (MAP) kinase pathways through the Unfolded Protein Response (UPR) signaling pathway driven by continued ER stress, which in turn could significantly affect the antiviral response of the COVID-19 patient [42,43,44]. The UPR response can be triggered by the overload of viral proteins such as S, E, and M, which are frequently found in the ER [45], although the latter is synthesized in the ERGIC compartment, where it recruits other viral proteins [46]. In addition, ORF8 was also found to induce ER stress, in this case through activation of inositol-requiring enzymes 1 (IRE1) and activation of transcription factor 6 (ATF6) branches of the ER stress pathway, while functioning as an IFN antagonist, consequently inhibiting interferon beta (IFN-β) production [47].

As predicted by Gordon et al., multiple interactions between viral and host proteins can also occur in the ER. In fact, approximately 40% of the entire virus–host interactome is associated with vesicle trafficking or endomembrane compartments [35]. One such viral protein, the SARS-CoV-2 M protein, has multiple putative binding partners in the ER, such as reticulon 4 (RTN4) or receptor expression-enhancing proteins 5 (REEP5) and 6 (REEP6). The former has an important role in ER morphology [48], whereas both REEP5 and REEP6 are members of the DP1 family of receptor accessory proteins that are expressed in ER and affect its cargo capacity of G Protein-Coupled Receptors (GPCRs) and, in turn, their surface expression [48,49,50]. It is worth mentioning that REEP proteins are associated with olfactory signaling pathways [51], and both REEP5 and REEP6 interact with the alpha subunit of the IL-8 receptor (CXCR1), and when the expression of these proteins is reduced, IL-8-stimulated cellular responses in these cells decrease, thereby reducing proper lung cell activity [52]. Interestingly, REEP5 expression in SARS-CoV-2 infected Caco-2 cells is downregulated, which has been proposed as a possible mechanism of olfactory dysfunction in patients with COVID-19 [51]. Although it is presumable that these interactions may alter ER morphology and IL-8-mediated responses, research on the direct effects of SARS-CoV-2 on them is currently limited.

Moreover, it has been described that nsp6 and ORF9c proteins interact with Sigma receptors, which are transmembrane proteins located in the ER and play several cellular functions, such as regulation of lipid biogenesis and remodeling, ER stress response pathways, inhibition of cytokine production, modulation of calcium signaling through the inositol trisphosphate receptor (IP3) and neuronal signaling, as well as cell differentiation, survival, and morphology, among others [53,54,55,56]. Both Sigma-1 (σ_1_) and Sigma-2 (σ_2_) receptors have similar pharmacological profiles and are found mainly in the central nervous system, although they can also be localized in other tissues. The former receptor is expressed mainly in neurons and oligodendrocytes of the hypothalamus, midbrain, olfactory bulb, and deeper laminae of the cortex, as well as in Purkinje cells of the cerebellum, adrenal glands, kidneys, testes, liver, lungs and heart, while the σ_2_ receptor is expressed in the hippocampus, cerebellum, motor cortex, and substantia nigra of the brain, as well as in the lungs, kidneys, and liver [53,57,58,59]. Consequently, alterations of both receptors by SARS-CoV-2 nsp6 and ORF9c proteins could explain some of the symptoms of COVID and long COVID diseases, such as affective and cognitive disorders, memory impairment, addiction, anxiety, suicidal ideation, depression, pain, and neurodegeneration, among others, making the pharmacological target of sigma receptors a candidate therapy to delay this clinical deterioration [35,60,61].

Thus, it has become clear that the virus could reconfigure the trafficking and structure of the ER through the interaction of its proteins with those of the host cell [35]. SARS-CoV-2 nsp7 protein, through its interactions with selenoprotein S (SELENOS), RAS oncogene family member 1 (RAB1A), mannosyl-oligosaccharide glycosidase (MOGS), cytochrome b5 reductase 3 (CYB5R3), and cytochrome b5 type B (CYB5B) [35], could alter the degradation of misfolded luminal endoplasmic reticulum (ER) proteins, vesicle trafficking from the endoplasmic reticulum to the Golgi apparatus and protein glycosylation patterns [62,63,64]. Other interactions described by Gordon et al. are those between SARS-CoV-2 ORF8 and some ER-related host proteins, such as endoplasmic reticulum lectin OS9 (OS9), lysyl oxidase (LOX), FKBP prolyl isomerase 7 (FKBP7), and ER degradation enhancer alpha-mannosidase-like protein 3 (EDEM3) [35]. Among them, OS9 and EDEM3, which are two ER-associated degradation-related (ERAD) proteins, play an important role in coronavirus-induced DMV formation [65,66].

In addition, coronavirus infection can also cause severe ER membrane restructuring as a consequence of double-membrane vesicle (DMV) formation during viral replication, as well as ER membrane exhaustion as a consequence of continuous viral particle synthesis [44]. Furthermore, as has been observed in other organelles that have undergone cytopathy following SARS-CoV-2 infection, the previous host cell status is a predictor of response to UPR. Among these predisposing factors, elderly COVID-19 patients with comorbidities tend to have a poor prognosis. Although this may be due to several causes, immunosenescence, genomic instability, telomeric attrition, and inflammasome formation could be involved in the increased pathophysiological responses to SARS-CoV-2 infection seen in elderly patients. Moreover, as oxidative damage is also known to be related to aging, such patients may be especially prone to ER stress [67].

### 2.3. Cytopathic Effects on the Golgi Apparatus

The Golgi apparatus is an organelle found in most eukaryotic cells. It consists of flattened stacks of membrane-bounded sacs called cisternae and plays a vital role in the processing, modification, and sorting of proteins and lipids as they move through the cell. It has also been described to be affected by SARS-CoV-2 infection, both by direct morphological and molecular alterations. Recent studies have shown that an extensive Golgi fragmentation can occur in infected lung epithelial cells, mainly triggered by SARS-CoV-2 S, M, E, nsp15, and ORF3a proteins, and its function and structure may be altered by SARS-CoV-2-induced upregulation of trans-Golgi network integral membrane protein 2 (TGN46) and downregulation of Golgi reassembly-stacking protein of 55 kDa (GRASP55) [68,69]. This fragmentation of the Golgi, which occurs following infection by several types of viruses, has been described to cause reduced expression of the major histocompatibility complex class I (MHC-I) as a direct consequence of defective membrane trafficking, which in turn could enhance viral replication [70]. Moreover, SARS-CoV-2-induced Golgi fragmentation has been proposed to be a predisposing factor for certain neurological manifestations of COVID-19 and long COVID diseases, such as an Alzheimer Disease-like phenotype and the aforementioned “brain fog”, which can also be triggered by fragmentation of this organelle [71]. Knowing that muscle, cartilage, nervous system, skin, and bone are some of the organs most sensitive to defects in Golgi-mediated membrane trafficking [70], it is possible that they could be the most affected by SARS-CoV-2-induced cytopathy within the Golgi apparatus.

As described by Gordon et al., SARS-CoV-2 proteins could also be able to interact with a wide repertoire of host proteins related to the Golgi apparatus [35]. One such interaction occurs between the M protein and host proteins such as YIF1A, FAM8A1, and certain members of the SLC30 family, which are expressed in the Golgi apparatus or ERGIC compartment and are involved in cellular functions such as vesicular transport, cellular zinc homeostasis, and membrane trafficking [72,73,74,75,76,77]. Similarly, ORF9c can interact with SLC30A6 [35], which has an important role in zinc homeostasis [78,79]. Therefore, it is presumable that the virus may alter these functions by interacting the M protein with these proteins, although further studies are required to determine the direct effects of these interactions. However, in the context of viral infection, zinc is known to enhance mucociliary clearance, decrease viral replication, enhance epithelial integrity, and attenuate inflammation, among other functions. In addition, it has been proposed to play an important role in antiviral immunity, and hypozincemia may be a predisposing factor for dysgeusia, which is a common symptom among COVID-19 patients [80,81]. Thus, studying the effects of ORF9c and M protein interactions with SLC30 family proteins to determine whether zinc imbalance occurs during SARS-CoV-2 infection, as well as considering these interactions as candidates for drug targets, could be an interesting strategy for the treatment of these zinc-related alterations.

Some other viral–host interactions predicted by Gordon et al. could alter the proper function and structure of the Golgi apparatus. SARS-CoV-2 nsp13 protein can interact with Golgi reassembly and stacking protein 1 (GORASP1), Golgin-A2 (GOLGA2), -A3 (GOLGA3) and -B1 (GOLGB1), phosphodiesterase 4D-interacting protein (PDE4DIP), GRIP and coiled-coil domain-containing proteins 1 (GCC1) and 2 (GCC2), and A-kinase anchoring protein 9 (AKAP9) [35]. These proteins, which are mainly localized in the Golgi apparatus, are responsible for the assembly and stacking of the cisternal membrane (GORASP1), the attachment of transport vesicles to the Golgi (Golgins) and protein kinase A (AKAP9) to the Golgi apparatus, as well as the organization of the subcompartments of the trans-Golgi network involved in membrane transport (GCC proteins) [82,83,84,85,86,87]. Moreover, host polysaccharide metabolism may be directly altered by some of these host–virus protein interactions, as the SARS-CoV-2 ORF8 protein has been described to interact with HS6ST2, CHPF, and CHPF2 [35], which are localized in the Golgi apparatus and are involved in glycosaminoglycan synthesis [88].

### 2.4. Cytopathic Effects on the Cytoskeleton and Plasma Membrane

Among all the structures present in eukaryotic cells, the cytoskeleton is elementary for the transport and organization of organelles and other subcellular compartments. It not only participates in most cellular biological processes and functions but also regulates them, in particular, cell–cell interactions, cell movement, and cell division [89]. This dynamic network is essentially composed of 3 filamentous protein polymers: actin filaments, microtubules, and centrioles that, together with other microtubule-associated proteins (MAPs), function as cellular scaffolds and, apparently, also as information processing and signaling systems. The central part of this structure is the centrosome or MTOC (Microtubule-Organizing Center), which is formed by two perpendicular cylinders immersed in an electronegative pericentriolar matrix. As known so far, all infectious viruses, such as those of the Coronaviridae family, penetrate and “hijack” host cell mechanisms to expand and proliferate, but SARS-CoV-2 seems especially prone to use the microtubule network and the MTOC for host cell infection, proliferation, and damage [90]. Thus, a profound remodeling of the cytoskeleton has been described in SARS-CoV-2-infected lung cells [68]. As described by Aminpour, M. et al., the centrioles of the MTOC can be considered as “the brain and the eye” for the rest of the cytoskeleton, as they are able to detect and respond to different electromagnetic signals in the visible and near-visible spectrum. There is evidence to suggest that SARS-CoV-2 is able to take control of this machinery by docking inside the centriole barrels of MTOCs and forming interactions between the negatively charged C-termini of microtubules and the positively charged spike protein of the virus, resulting in modulation and disruption of MTOC and microtubule function at different levels [90].

When SARS-CoV-2 attacks a host cell, it binds the spike protein to the membrane ACE2 receptor, which is associated with and anchored to the microtubules of the cytoskeleton, in particular with β-tubulin. The virus then enters the cell and is able to move to specific locations along the cytoskeleton structures thanks to host cell motor proteins. SARS-CoV-2 rearranges these structures to use them as signals or displace them when they pose an obstacle [90]. In addition, SARS-CoV-2 can also control its local spread by inducing the formation of filopodia in infected host cells to contact surrounding cells. This control is thought to be achieved by manipulating or activating certain kinases [90]. In this context, it should be noted that some immune cells, such as alveolar macrophages, are also target cells of SARS-CoV-2 infection. In these immune cells, MTOC is usually behind the nucleus playing a role of directional guidance and cell movement, but when interacting with antigen-containing cells, forming an immune synapse, MTOC moves to the front and is directly involved in the release of cytokines, interleukins, and tumor necrosis factor (TNF), among others. If SARS-CoV-2 takes control of MTOC, this release is uncontrolled, leading to a cytokine storm, an excessive inflammatory response that causes extensive cellular damage [90]. In this way, the virus manages its own intracellular transport and replication, cell-to-cell spread by inducing filopodia formation, and immune system malfunction ranging from hyperinflammatory cytokine storm to a poor response [91].

In addition to the interactions with the cytoskeleton detailed above, Gordon et al. described that four other viral proteins, ORF9b, ORF10, nsp11, and nsp13, could interact with host proteins responsible for the integrity of the cytoskeleton [35], which could alter microtubule dynamics and cell polarity within infected cells. One such protein is Radixin (RDX), which is involved in connecting plasma membrane proteins to the cytoskeleton [92] and has been described to interact with the SARS-CoV-2 nsp13 protein [35]. The SARS-CoV-2 ORF9b protein could do so by interacting with the host protein MARK2 (Microtubule Affinity Regulatory Kinase 2) [35]. The MARK2 protein is involved in the regulation of microtubule dynamics, which in turn may affect cell migration [93]. Microtubules are important for cell migration as they provide the structural framework for the formation of the cytoskeleton, which is necessary for cell movement. MARK2 protein phosphorylates microtubules, which may affect their stability and organization [94]. In fact, it is known that the activity of this protein could be inhibited by the formation of complexes [95] suggesting that its interaction with ORF9b of SARS-CoV-2 could also alter its function. Furthermore, as this viral protein can also interact with the SLC9A3 regulator 1 (SLC9A3R1) and the cold shock domain containing E1 (CSDE1) [35], membrane–cytoskeletal protein binding and the formation of microvilli and stress granules could be promoted through the interaction of ORF9b with these proteins [96,97].

Moreover, nsp11 has been found to interact with tubulin folding cofactor A (TBCA) [35], which regulates host cell microtubule stability and tubulin availability [98,99]. This interaction could also lead to changes in microtubule stability by altering the capture and stabilization of beta-tubulin intermediates, which, consequently, could affect the cell’s ability to maintain proper structure and function. Finally, SARS-CoV-2 nsp13 interacts with the host protein ninein-like (NINL) [35]. This protein is essential for mitosis, as it promotes microtubule nucleation and regulates cytokinesis and chromosome segregation [98]. In addition, some studies have suggested that NINL can be associated with cilia and, when altered, may be involved in the development of ciliopathies [99,100,101].

The nsp3 protein has also been described to interact with other proteins of the centrosomal complex, such as ninein (NIN), pericentrin (PCNT), centriolin (CNTRL), centromere protein F (CENPF), and some members of the centrosomal protein family (CEP) [35]. Furthermore, in the context of viral replication, it is noteworthy that SARS-CoV-2 ORF10 can cause ciliary dysfunction through its interaction with certain members of the cullin-2 complex (CUL2), causing downregulation and loss of stability of some host proteins related to ciliogenesis [35,102]. 

Knowing that most of the host proteins detailed above are responsible for cilia formation and may be involved in the development of ciliopathies when altered [103], it is presumable that the virus could take advantage of these interactions with the centrosomal complex to replicate. Once inside the cell, the virus disrupts the normal functioning of the cilia by interfering with the microtubules that form the ciliated structure. This can result in the shortening or loss of cilia, which in turn would have a number of effects on organs in which hair cells play a key role, such as the respiratory system. Consequently, this could create an environment in which the virus can easily replicate, as cilia are responsible for removing pathogens from the respiratory tract. Without functional cilia, the virus manages to establish itself in the respiratory tract and replicate. In addition, alterations in the cilia could contribute to short- and long-term symptoms of respiratory dysfunctions, such as anosmia, respiratory problems caused by increased mucus production, or chronic cough due to poor mucus and pathogen clearance, among others [104].

To date, little evidence of cell membrane alterations in long COVID pathology has been described. However, it is possible that such alterations modify, during the course of COVID-19 pathogenesis, the functionality of those cells whose activity takes place primarily at the plasma membrane, such as cells that establish neuronal or immune synapses. In this regard, Gordon et al. described that the SARS-CoV-2 ORF3b protein interacts with Stomatin-like 2 (STOML2) [35], whose dysfunction has been linked to altered formation of the T cell receptor (TCR) signaling complex [105]. Interestingly, a recent study showed that the SARS-CoV-2 spike protein could directly suppress immune synapse (IS) formation in CD8^+^ T cells, which could be used by the virus as a way to evade the cytotoxicity response against infected cells. Since activated T cells express the ACE2 receptor, this would, in turn, facilitate the entry of the SARS-CoV-2 virus, and when the cell is infected, the S protein is targeted to the IS [106]. This sequestration tactic is also used by other viruses, such as HIV-1 [107]. Outside the immune system, alterations of GPCRs have been described following SARS-CoV-2 infection in the lung. By hijacking GPCR signaling pathways, the virus may enhance replication and induce secondary effects on pulmonary ionic balance [108]. Furthermore, knowing that the nsp10 protein can interact with the mu subunit of the AP-2 complex (AP2M1) [35], which is involved in clathrin-dependent endocytosis and autophagy-induced claudin 2 (CLDN2) degradation via endocytosis [109], it is presumable that nsp10-AP2M1 could inhibit the formation of this complex, thus preventing the formation of clathrin-coated vesicles. Moreover, proper cycles of phosphorylation and dephosphorylation are important for both endocytosis and synaptic vesicle recycling [110], and AP2M1 phosphorylation has been reported to be induced after SARS-CoV-2 infection in H522 cells, whereas its presence has been shown to be crucial in viral infection in those cells expressing low levels of TMPRSS2 [111,112]. Although the role of AP2M1 in coronavirus replication remains unknown [111], it is possible that disruption of its proper function by SARS-CoV-2 infection may alter cellular communication and neuronal activity.

As for cytopathy in the cytoskeleton and related cellular processes, its alterations caused by SARS-CoV-2 infection could contribute to the development of prolonged COVID through several mechanisms. For example, the altered organization of the cytoskeleton could disrupt the cellular transport of important molecules within the cell, such as neurotransmitters or signaling molecules, which could contribute to the persistence of some neurological symptoms such as fatigue, “brain fog”, and loss of smell or taste, among others. Dysregulation of the immune response, including cytokine storming resulting from MTOC dysfunction, may contribute to ongoing inflammation and tissue damage. In addition, alterations in microtubule dynamics could contribute to the development of chronic lung damage as a consequence of cilia dysregulation [113], as well as other organ dysfunctions observed in long COVID.

### 2.5. Cytopathic Effects on the Nucleus

The nucleus can also undergo morphological and functional alterations when the cell is infected by SARS-CoV-2. In fact, some viral proteins alone cause some of these changes. On the one hand, it has been widely described that some of the SARS-CoV-2 proteins block interferon-mediated responses in the host cell through different mechanisms [114], some of which are due to an alteration of nuclear-cytoplasmic transport. ORF6 expression subverts the nucleoporins RAE1 and NUP98, as this viral protein is known to interact with nuclear pore proteins, consequently altering nuclear import. In addition, it can also cause a reduction in nuclear size and disruption of cell growth if overexpressed [115]. It has also been described that the nsp9 protein of SARS-CoV-2 interacts with some proteins of the nuclear pore complex, such as NUP54, NUP58, NUP62, NUP88, and NUP214 [35]. ORF6, as well as ORF3b, block IFN signaling by inhibiting the entry of transcriptional factors such as STAT into the nucleus, thereby impairing the induction of transcription of IFN-stimulated genes (ISGs) and, consequently, overcoming antiviral responses [116]. Similarly, SARS-CoV-2 N protein expression also blocks the expression of ISGs, in this case through inhibition of STAT1 and STAT2 phosphorylation, which is a key step in their translocation to the nucleus [117]. The nsp12 protein does this by inhibiting the nuclear translocation of IRF3 [114], which is an interferon regulatory transcription factor with an important role in the antiviral response [118]. On the other hand, SARS-CoV-2 can inhibit mRNA export from the host nucleus, presumably through binding of nsp1 to nuclear RNA export factor 1 (NXF1), thereby suppressing host cell gene expression [119] (Table 1).

Other morphological alterations in the nuclei of infected cells have also been described. Post-mortem histopathological analysis of salivary glands from patients with COVID-19 revealed that some of the cells of the duct lining epithelium might undergo nuclear pleomorphism and vacuolization of their cytoplasm and nucleus, whereas acinar cells showed enlarged nuclei [120]. Similar results were observed in keratinocytes of the gingival junctional epithelium [120]. Histopathological examinations of the lungs of patients who died of COVID-19 also revealed alterations in the nuclear morphology of some cells, mainly in type II pneumocytes, which often show large nuclei and prominent nucleoli [121]. It is also noteworthy that fusion between infected and healthy cells can occur, leading to the formation of multinucleated syncytia, especially in type II pneumocytes [122]. This, together with the long-term persistence of viral RNA and the occurrence of thrombosis, are three of the main hallmarks of advanced COVID-19 disease [123].

Recent studies have reported that folate-mediated monocarbon metabolism (FOCM) is altered in some patients with long COVID. This is a metabolic network occurring in the nucleus, mitochondria, and cytoplasm, which can be stressed during the viremia phase of SARS-CoV-2 replication, leading to serine and glutathione depletion, increased oxidative stress, and altered methyl group delivery mechanisms [124]. S-adenosylmethionine (SAMe), a co-substrate involved in methyl group transfer, plays a central role in this metabolic pathway and is transported to the cell nucleus through nuclear pores [124]. Therefore, knowing that interactions of certain SARS-CoV-2 with host cell nuclear pores alter the nuclear import of certain molecules, one explanation for the development of FOCM in these infected cells could be that the nuclear transport of SAMe may also be altered. In addition, the SARS-CoV-2 nsp9 protein is known to interact with methionine adenosyltransferase 2B (MAT2B), which is responsible for SAMe biosynthesis from methionine and ATP [125], which could also alter this metabolic pathway. Consistent with the aforementioned cytopathic effects on other cellular organelles, the persistence of viral proteins within infected cells or side effects of their interactions with host proteins could also alter nuclear morphology and impair nuclear trafficking, ultimately leading to an attenuated long-term antiviral response and altered gene expression. However, there is little evidence for the possible implications of nuclear cytopathies caused by SARS-CoV-2 in the development of long COVID disease.

## 3. Direct Cytopathic Effects on Various Tissues

### 3.1. Cytopathic Effects on the Central Nervous System

Early in its spread, SARS-CoV-2 was identified as a virus capable of infecting the central nervous system (CNS), as several COVID-19 patients suffered neurological manifestations followed by infection [126] Since then, several possible routes for the spread of the virus to the CNS have been proposed. In addition, it has been accepted that the neurological manifestations observed in COVID-19 patients may be due to direct CNS infection and/or due to side effects of systemic infection, such as hypoxemia or cytokine release syndrome [127]. Consequently, given that the CNS is frequently affected by SARS-CoV-2 infection, it is not surprising that neuropsychiatric symptoms are common in patients with prolonged COVID.

The CNS has several strongholds that protect it against most viral infections: the external multilayer barriers, the blood–brain barrier (BBB), and the immune system. However, some viruses enter the CNS by retrograde hematogenous or neuronal routes, leading to debilitating direct immune-mediated pathologies [128]. SARS-CoV-2 virus enters the nervous system by direct infection of tissue nerve endings and/or by utilizing the axonal transport machinery, as well as entering the CNS directly through the blood–brain barrier [127]. Thus, it uses transmucosal olfactory invasion as a gateway to access the central nervous system, which could be a consequence of ACE2 expression in the olfactory epithelium, specifically in sustentacular cells, Bowman’s gland, and horizontal basal cells [129]. In fact, SARS-CoV-2 penetrates the olfactory bulb by infecting its epithelium instead of the sensory neurons, leaving a trail of viral RNA as the infection progresses on its way to the CNS [126,130]. Infection can also occur directly in the endothelial cells of the BBB, whose tight junctions are disrupted, thus altering permeability (Figure 2) [129].

In addition, it should be noted that coronaviruses are capable of infecting macrophages, astroglia, and microglia. However, glial cells generate an immune response to SARS-CoV-2 infection by secreting proinflammatory factors (IL-6, IL-12, IL-15, and TNF-α) that mediate inflammation and cell damage in the nervous system [131]. In addition, elevated levels of proinflammatory and anti-inflammatory cytokines have been described in cerebrospinal fluid in COVID-19 patients with neurological manifestations [132]. Among these, high levels of IL-6 and IL-8 are thought to play an important role in BBB dysfunction. Furthermore, IL-6 was experimentally shown to alter tight junctions of endothelial cells, and, on the other hand, IL-8 in CNS is produced by microglia and endothelial cells and has an important role in leukocyte chemotaxis to the CNS [129]. Thus, the cytokine storm may cause dysfunction of the BBB, thereby altering its intrinsic ability to block both immune cell and pathogen infiltration into the CNS and, consequently, making possible the transition to a neuroinflammatory state that may cause CNS damage [129]. After bypassing physical barriers, SARS-CoV-2 could infect the CNS. This could take place mainly in neurons and glial cells expressing the hACE2 and/or neuropilin-1 receptor [8], which are widespread throughout the brain [133]. Interestingly, olfactory tubercles, paraolfactory gyrus, and olfactory epithelium express elevated levels of neuropilin-1, so their direct infection by SARS-CoV-2 may explain some of the neurological manifestations [134].

The clinical neurological manifestations vary in severity from headache to syncope, anoxic seizures, stroke, and encephalitis. These can be explained, in principle, by hypoxia, anaerobic metabolism in the cells of the central nervous system, as well as the formation of cellular and interstitial edema, ischemia, and vasodilatation in the cerebral circulation. Neurological symptoms have also been frequently reported in patients with COVID-19, such as anosmia, hypogeusia (or ageusia), confusion, seizures, and encephalopathy [135]. In some cases, endothelial ruptures may occur in cerebral capillaries, causing CNS hemorrhages, which worsen the course of the disease [126]. In addition, viral encephalitis and some opportunistic infections can occur in the brain of some patients with COVID-19, especially in those with lymphopenia [136]. 

Although the acute effects of SARS-CoV-2 infection on the nervous system have been widely described previously, the molecular mechanisms underlying the long-term effects are less well described. These include complications such as post-traumatic stress disorder, depression or anxiety, memory problems, insomnia, sleep disorders, cognitive impairment, impaired concentration, headaches, muscle weakness, dizziness, critical illness neuropathy, residual olfactory disorder, and acute inflammatory polyradiculopathy [137,138]. In addition, neurological complications of prolonged COVID are observed in the CNS and peripheral with diseases such as encephalitis, myelitis, myositis, Guillain Barré syndromes, and cognitive impairment [139]. Prolonged COVID may be the result of persistent neuroinflammation triggered during acute infection or other types of autoimmune-related changes. However, there is currently a lack of clear evidence to support either hypothesis [140].

Interestingly, Villadiego et al. studied, in a murine model, neuronal death caused by SARS-CoV-2 infection in the cortex, hippocampus, and hypothalamus, which are brain regions susceptible to allowing high viral replication. In this study, they observed that this virus induces neuronal apoptosis through Caspase-3, especially in the dentate gyrus of the hippocampus. In addition, previous studies have reported that ventral brain blood vessels may undergo structural alterations in some patients with COVID-19, and morphological changes in microglia were also described. The latter, which are the result of microglial activation, are characterized by a retraction of the projections and an enlargement of the cell body, while alterations in cerebral blood vessels may appear as a consequence of inflammatory activation of their permeability, which is also observed in other pathologies such as multiple sclerosis [127]. Regarding apoptosis in the dentate gyrus of the hippocampus, it has been previously described that certain infections affecting this region, such as bacterial meningitis, can also induce apoptosis, and learning deficits often occur in survivors [141]. Thus, since this brain region is key in spatial navigation and memory formation, the persistence of memory problems and cognitive impairment in long COVID could depend on the severity of cytopathic damage in the hippocampus. Moreover, the presence of alterations in cerebral blood vessels could be explained by direct infection by SARS-CoV-2 [128]. This was previously described in two different animal models, in which they observed that the SARS-CoV-2 nsp5 protein induces the cleavage of the essential modulator of NF-κB (NEMO) in infected endothelial cells, inactivating it. Interestingly, inactivating mutations in the NEMO gene can cause encephalopathy, seizures, and stroke, which could explain the presence of these symptoms in patients with COVID-19 [142].

Some other regions of the nervous system may also be affected by SARS-CoV-2 infection. For example, the pituitary and adrenal glands contain cells that express ACE2, making them potential targets of the virus [143]. In fact, it has been hypothesized that chronic fatigue syndrome, which is one of the most recurrent symptoms among patients with prolonged COVID, could be triggered by endocrine dysfunctions due to an alteration in the hypothalamic–pituitary–adrenal (HPA) axis [144]. Knowing that the virus infects these three elements of the HPA axis, it is presumable that long-term cytopathy or cell death followed by direct viral infection could contribute to the development of persistent chronic fatigue. Another brain region that can be directly affected by the virus is the limbic system, which may undergo volume loss and microstructural changes in the white matter [145]. Outside the CNS, it should be noted that SARS-CoV-2 can cause neuromuscular alterations, which have an impact on motor control and lead to muscle fatigue. In this context, neuronal demyelination has been described in patients infected by other coronaviruses, such as SARS-CoV, which could explain the muscle fatigue [142].

In addition, it is known that some of the peripheral cytokines produced as a consequence of SARS-CoV-2 infection in other tissues trigger astrocytes and microglia, which in turn produce neural cytokines that may participate in the development of neuroinflammation. Thus, the establishment of a neuroinflammatory state could contribute to the alteration of certain brain circuits, such as GABAergic transmissions, which are reported to be altered in some patients with COVID-19 and may contribute to their chronic fatigue [146]. Thus, it is presumable that long-term neurological sequelae following COVID-19 also goes beyond direct CNS infection.

### 3.2. Cytopathic Effects on the Respiratory System

As a respiratory pathogen, it is well known that the main target of the SARS-CoV-2 virus is the respiratory system, although it has been reported that it can affect most human tissues [147]. Therefore, cells in contact with the external environment that express viral receptors on their surface, such as ACE2 or NRP1 [9], and entry cofactors such as TMPRSS2, which are highly abundant in the respiratory tract, are the initial target cells of SARS-CoV-2 infection, and nasopharyngeal-associated lymphoid tissue (NALT) is the primary first lymphoepithelial barrier controlling SARS-CoV-2 entry [148]. Within the respiratory tract, the conducting airways and gas exchange regions of the lung are the main targets of the virus, primarily affecting the ciliated and secretory cells of the human airway epithelium, as well as type II alveolar cells. The cytopathy affecting these cells is characterized by the development of plaques, shedding and internalization of cilia, cell death, damage to the epithelial barrier, and detachment of infected ciliated cells containing a viral reservoir, among others [113,149,150,151]. As a consequence of the extensive damage that can occur during viral replication, a loss of mucociliary clearance may occur, along with dampened cytokine responses [151].

Nasal ciliated and goblet nasal epithelial cells, microwells, dendritic cells (DCs), macrophages, and T and B lymphocytes can be found in the upper airways, among other cells [152]. It has been shown that, among the diversity of cells that are part of the upper respiratory epithelium, the multiciliated cells of the nasal respiratory epithelium are the main targets for SARS-CoV-2 entry and replication [153]. SARS-CoV-2 causes the dedifferentiation of multiciliated cells through the negative post-transcriptional regulation of the transcription factor FOXJ1, which is a protein with an important role in orchestrating the production of motile cilia, thus inducing a rapid loss of the ciliary layer, altering mucociliary clearance, and causing the death of multiciliated cells, which consequently promotes the infection of the respiratory tree and increases the risk of secondary infections in patients with COVID-19 [113,154]. In the lower respiratory tract, findings in post-mortem tissue and experimental models indicate diffuse alveolar damage, hyaline membrane formation, pneumocyte desquamation, presence of platelet-fibrin thrombi in small arterial vessels, atypical pneumocyte hyperplasia, accumulation of protein exudate and fibrin in alveolar spaces, formation of multinucleated cells, and dead cells due to pyroptosis and apoptosis [121,149,155].

Type II pneumocytes infected with SARS-CoV-2 may present protein accumulation, alterations in the morphology of their mitochondria, and distended endoplasmic reticulum cisterns [155], and the presence of translucent lipid droplets of homogeneous appearance, without surrounding membrane and with attached mitochondria can also be observed [121]. It has also been described that once SARS-CoV-2 invades the alveoli, it can be taken up by local alveolar macrophages, subsequently producing proinflammatory cytokines such as type I interferon, which act on neighboring type II alveolar pneumocytes by activating aryl hydrocarbon receptors (AhR), which consequently translocate to the nucleus promoting the production of mucus that accumulates in the alveoli. The accumulation of mucus causes a gradual decrease in O_2_ and CO_2_ exchange, which in turn leads to hypoxia [156].

There are two populations of macrophages in the lung: alveolar macrophages, which can be found near type I and II pneumocytes, and interstitial macrophages, which are found between the alveolar epithelium and the microvascular endothelium [157]. These macrophages express ACE2, TMPRSS2, and Furin [158,159], thus being susceptible to SARS-CoV-2 infection [158]. Although this infection is abortive in monocyte-derived macrophages (MDM) and dendritic cells (MDDC), it can induce the production of a wide repertoire of pro-inflammatory cytokines, such as IL-6 or TNF-α [160]. In fact, it has been observed that increased production of the latter two cytokines, together with impaired response to IFN-I and persistence of viral load in the blood, are also some of the characteristics of severe COVID-19 [161]. However, it is still unclear whether respiratory tract macrophages are the main source of proinflammatory cytokines following SARS-CoV-2 infection [162].

These cytopathic effects caused by SARS-CoV-2 infection at the cellular level have contributed to the development of post-acute COVID-19 [24], which is defined as the persistence of symptoms and/or long-term complications beyond 4 weeks from the onset of symptoms. At the respiratory system level, the predominant symptoms are dyspnea, fatigue, cough, and persistent need for oxygen [24,163]. In addition, pulmonary fibrosis has been described as a possible long-term complication, and it has been hypothesized that pulmonary vascular disorders may be responsible for long-term respiratory symptomatology. Respiratory disturbances have been correlated with dysregulated iron metabolism, and it has also been reported that patients with pulmonary sequelae often have metabolic abnormalities related to pulmonary repair and fibrosis [164]. In addition, basal cells of the airway epithelium often avoid SARS-CoV-2 infection and, in turn, enable the lung to restore damage caused by viral replication. Previous studies have observed that these cells exhibit migratory behavior suggesting their active role in respiratory epithelial repair, along with cilia regeneration by the remaining epithelial cells, although the regenerative power is often incomplete and dysregulated, which in turn could lead to the development of respiratory symptoms in long COVID patients [113,154].

### 3.3. Cytopathic Effects on the Circulatory System

SARS-CoV-2 viral particles can infect endothelial cells, as these express both ACE2 and TMRPSS2. In addition, heparan sulfate has also been reported to enhance virus binding, as well as gangliosides and sialic acid-containing glycoproteins, which localize in endothelial cell membranes [165,166,167]. The marker CD147, which is expressed on endothelial cells, has also been hypothesized to serve as an entry receptor for SARS-CoV-2, although data regarding its role as an alternative direct receptor remain unclear [168]. In addition, the proprotein convertase furin, which is also present in endothelial cells, can preactivate SARS-CoV-2 particles during infection, and when a conserved C-terminus of the spike protein is exposed after cleavage of the furin-like cleavage sites on this protein, it binds to the receptors Neuropilin-1 and Neuropilin-2, which are also present on these cells, consequently increasing infectivity [9,165]. Infection occurs through transcellular movement due to the close contact of the vascular endothelium with the epithelial cells of the lung. In addition, the virus can also create a paracellular invasion route causing significant changes in permeability by passing through the interstitial space, adhering to the αVβ3 integrin expressed abluminal to the vascular endothelium and in almost all tissues and cells of mesenchymal origin, thus promoting the loss of the integrity of the endothelial barrier and consequently facilitating virus dissemination since the change in endothelial morphology is probably accompanied by the disruption of intercellular junctions. Consequently, the endothelium loses its property of being a complete monolayer, causing a detachment of endothelial cells and loss of barrier integrity [169]. In fact, SARS-CoV-2 infection of endothelial cells has been documented to induce endothelial dysfunction, despite being a non-productive infection [154]. This endothelial dysfunction is a consequence of several factors, such as the fact that SARS-CoV-2 protein S negatively regulates ACE2 expression, thus altering the renin–angiotensin–aldosterone system (RAAS) and the bradykinin–kallikrein pathway [24]. This favors the accumulation of angiotensin II, which increases endothelial permeability, thus allowing the movement of the virus from the circulation to pericytes, which are cells of the vascular wall of the basement membrane of blood microvessels, and exhibit direct contacts with the endothelium mediated by platelet-derived growth factor receptor beta (PDGFR-β) and angiopoietin I (Angpt I), which in turn interact, respectively, with PDGF-β and Angpt II in endothelial cells. These interactions are altered during SARS-CoV-2 infection, leading to a loss of endothelial integrity, thus increasing thrombogenic exposure of the basement membrane, which in turn could induce hypercoagulation, hypofibrinolysis, and prothrombotic states [170]. Furthermore, downregulation of ACE2 by SARS-CoV-2 infection impacts mitochondrial function and morphology by increasing mitochondrial ROS production and releasing mitochondrial DNA (mtDNA). The latter activates TLR9, which triggers inflammatory responses mediated by NF-κB-derived proinflammatory cytokine synthesis that compromise endothelial cell function by decreasing nitric oxide synthase (eNOS) expression, thereby impairing nitric oxide (NO) bioavailability. Interestingly, spike protein alone can downregulate ACE2 and eNOS in endothelial cells while increasing their glycolytic activity [171,172].

In addition to the endothelial infections described above, the virus can also directly affect cardiac cells such as cardiomyocytes, interstitial cells, and macrophages that invade cardiac tissue. Interestingly, direct infection of cardiomyocytes via ACE2 has been described to promote contractile deficits, sarcomere disassembly, cell death, and cytokine production. In addition, several pathophysiological studies of the heart of deceased COVID-19 patients reported myocardial hypertrophy and necrosis, focal myocardial fibrosis, lymphocytic inflammation, invasion of mononuclear inflammatory cells into the myocardial interstitium, small vessel coronary artery disease, acute myocardial infarction, and interstitial edema, among other less common events [173,174].

SARS-CoV-2 infection also causes elevation of cardiac biomarkers, electrocardiographic and myocardial abnormalities, and cardiac arrhythmias, as well as severe complications such as acute coronary syndrome due to plaque rupture or thrombus, an imbalance between oxygen supply and demand, myocardial injury due to disseminated intravascular coagulation, nonischemic myocarditis-type injury, stress-induced cardiomyopathy, and cytokine release syndrome. The latter is responsible for an intense release of multiple cytokines and chemokines [175]. The proinflammatory cytokines IL-1, IL-6, IFN-γ, and TNF-α depress myocardial function through activation of the neural sphingomyelinase pathway through reduction of nitric oxide-mediated beta-adrenergic signaling [175]. In accordance with these cytopathic effects at the level of the circulatory system, it has been reported that, among the symptoms that persist in long COVID, some cardiovascular events are frequent, such as arrhythmias, cardiac lesions, chest pain, palpitations, hypotension, increased heart rate, venous and arterial thromboembolic diseases, myocarditis, and acute heart failure [24,176,177], which could contribute to increased patient morbidity and mortality. In addition, increased cardiometabolic demand may be a persistent sequela in some recovered patients, which in turn can be related to dysregulation of the RAAS system [24]. Some studies suggest that possible mechanisms for the development of these symptoms include persistent damage by the direct viral invasion of cardiomyocytes and subsequent cell death, infection, and inflammation of endothelial cells, transcriptional alteration of multiple cardiac cell types, activation of the complement system, complement-mediated coagulopathy and microangiopathy, downregulation of ACE2, autonomic dysfunction, elevated levels of proinflammatory cytokines, and activation of TGF-β signaling through the Smad-signaling pathway to induce subsequent fibrosis and scarring of cardiac tissue [21,144,178,179,180]. 

Persistence of immune responses followed by infection, as well as a maintained presence of the virus in immunoprivileged sites and the establishment of an autoimmune state, have also been hypothesized as a cause of some of the extrapulmonary post-acute sequelae of COVID-19, such as the given in the circulatory system [164,178,179,181]. Regarding the development of autoimmune disorders, it is worth mentioning that the development of antiphospholipid antibodies has been reported in some patients with COVID-19 in the acute phase, in which the development of thrombotic complications and vascular inflammation may occur [164]. Finally, retro-integration of SARS-CoV-2 into the genome of infected human cells, and the expression of chimeric transcripts containing viral and cellular sequences, have also been proposed as a mechanism for the continued activation of the immune–inflammatory–procoagulant cascade, which could explain the range of post-acute COVID-19 cardiovascular sequelae [181].

### 3.4. Cytopathic Effects on the Immune System

The initiation of the immune response begins with the activation of pattern recognition receptors (PRRs), which occurs when specific pathogen-associated molecular patterns (PAMPs) are recognized. When PAMPs expressed by host cells, such as viral material, are recognized by PRRs, an innate immune response is triggered, leading to the release of cytokines. In the case of coronaviruses, receptors such as Toll-like receptor 7 (TLR7), RIG-1, and MDA 5 [182] activate signaling cascades leading to the expression of type I IFN (α and β) and other cytokines aimed at suppressing viral replication. Both adaptive and innate immune responses are key in viral clearance when they occur in a coordinated manner; however, due to genetic alterations or acquired disorders, a deficient response may develop [182,183].

The role of NK cells and cytotoxic (CD8^+^) T cells in the context of healthy pathogen clearance is noteworthy. On the one hand, viral antigens can be processed by infected cells through MHC-I and consequently presented to the CD8^+^ T cell receptor (TCR), inducing the release of proteolytic enzymes and, in turn, generating cytotoxicity, with subsequent elimination of virus-infected cells. On the other hand, NK cells develop a coordination of innate and adaptive immunity leading to the direct destruction of infected cells and regulation of the inflammatory response [182]. In addition, SARS-CoV-2 can weaken the T cell response by downregulation of MHC class I and II molecules [184]. In the case of MHC-I, it has been previously described that the SARS-CoV-2 ORF8 protein downregulates its expression through the Beclin 1 mediated autophagy pathway [185].

In addition, it has been widely reported that patients with severe COVID-19 disease often have severe lymphopenia (≤600 cells/mm^3^), with low levels of CD4^+^ and CD8^+^ T cells, NK cells, and regulatory T cells (T_reg_), while exhibiting higher levels of macrophages and monocytes [182]. An increased involvement of cytotoxic T lymphocytes has also been observed in the acute phase. In addition to the above, due to various congenital or acquired conditions, patients with severe COVID-19 often have difficulty in clearing the viral infection, which favors an exacerbated immune response. This is one of the main triggers of the so-called cytokine release syndrome [186]. Moreover, when coinfection occurs in advanced stages, there is an increase in the neutrophil count, as well as an increase in the neutrophil/lymphocyte ratio, indicating increased disease severity and an unfavorable prognosis. It is also important to note that secondary lymphoid organs can also be affected by SARS-CoV-2 infection. It has been reported that lymph nodes may undergo atrophy or even necrosis, as well as the spleen. The spleen may also show macrophage proliferation and apoptosis, hemorrhagic necrosis, and splenic cell degeneration [182].

Furthermore, it appears that a burst of type I IFN in the early stages of COVID-19 disease leads to protection, whereas a delay in interferon production results in an inability to control viral replication, which consequently leads to cellular damage of airway epithelia and lung parenchyma and, ultimately, to a lethal storm of inflammatory cytokines. Interestingly, the SARS-CoV-2 proteins ORF6, ORF8, and N are potential inhibitors of the type I interferon signaling pathway. These three viral proteins have a strong ability to inhibit virus replication, and they strongly inhibit type I interferon (IFN-β) and the NF-κB-sensitive promoter [187]. In addition, previous studies reported that the SARS-CoV M protein could block the formation of TRAF3, TANK, and TBK1/IKK complexes, consequently, inhibiting type I IFN production, which could also be triggered by SARS-CoV-2 infection [184].

The cytokine storm can lead to activation of the entire immune system or multiorgan failure, among other symptoms, although it is potentially lethal. This occurs in severe cases, where the exacerbated immune response triggered by SARS-CoV-2 infection is associated with respiratory system dysfunction [188]. This hypercytokinemia is initiated by dysregulated synthesis of proinflammatory mediators such as IL-1B (neutrophil activation and endogenous pyrogen), IL-6 (neutrophil activation), IL-7 (T lymphocyte differentiation), IL-8 (neutrophil activation), IL-9 (lymphocyte growth factor), IL-10 (suppresses lymphocyte proliferation and cytokine production), and TNF-α (activates neutrophil response and increases C-reactive protein synthesis), among other cytokines [189]. Some of these severe COVID-19 patients may suffer from Macrophage Activation Syndrome (MAS) or Hemophagocytic Lymphohistiocytosis (sHLH). They are also related to alterations in CD8^+^ T cell and NK cell activity and can cause symptoms such as fever, lymphadenopathy, anemia, coagulation, elevated serum ferritin and triglyceride levels, liver dysfunction, splenomegaly, and multi-organ failure [190]. Furthermore, one study reported that SARS-CoV-2 hijacks the host metabolism in some patients, including a decrease in tricarboxylic acid cycle metabolites and alterations of purine, pyrimidine, arginine, and tryptophan metabolisms, all of which correlate with proinflammatory cytokine production in these patients [191]. 

Regarding the direct effect of SARS-CoV-2 on the host cell, it should be noted that infected cells can undergo pyroptosis, a type of programmed lytic cell death that usually occurs after infection by the cytopathy virus, in which activation of caspase 1 stimulates the release of damage-associated molecular patterns (DAMPS). When these DAMPS are recognized by certain neighboring cells, such as alveolar macrophages, epithelial cells, and endothelial cells, proinflammatory cytokines and chemokines are produced. Some of the most relevant cytokines in this scenario are IL-6, IP-10, macrophage inflammatory protein 1α (MIP1α), MIP1β, and MCP1, all of which attract monocytes, macrophages, and T cells to the site of infection. Consequently, this cell recruitment can promote inflammation and, in turn, establish a proinflammatory feedback loop that ultimately damages lung tissue [192]. 

Furthermore, the immune response to SARS-CoV-2 itself could be responsible for the generation of long-term symptomatology after COVID-19 infection, possibly by triggering an ongoing inflammatory process due to an aberrant humoral or cellular response [193]. Possible immunopathological mechanisms that could explain this effect include the continued presence of viral reservoirs, mitochondrial dysfunctions, alterations of the microbiome, disorders of immunometabolism, cross-reactivity of antibodies against SARS-CoV-2 proteins with host proteins, alterations of the RAAS system, and a delay in viral clearance, among others [181]. It has been previously discussed that a strong IL-6-mediated response could impair recovery after SARS-CoV-2 infection [190]. In fact, this cytokine, along with others such as TNF-α and IL-1β, remain elevated in patients with long COVID [193], which could be explained by the high number of monocytes and macrophages usually observed in patients with COVID-19 [182].

However, despite the large amount of research on COVID-19, relatively little is known about the molecular underpinnings of these long-term effects, which are still under investigation. Although immunologic differences between patients who develop long COVID and those with a rapid recovery are still being studied, one study showed that CD8^+^ T cell responses were lower in patients with long COVID, and the number of CD4^+^ T cells producing IFN-γ was low in hospitalized patients with advanced age [194]. Furthermore, profound alterations in many immune cell types have been shown to persist for weeks or even months after SARS-CoV-2 infection, depending on the dynamics of the T cell immune landscape, integrated with patient-reported symptoms. Furthermore, the alterations occurring among T cell subsets show different dynamics directly dependent on time and severity, which, in severely convalescent patients, translates into a state of CD4^+^ and CD8^+^ T cell depletion/senescence, as well as perturbations in CD4^+^ T_reg_ cells. In particular, CD8^+^ T cells show a high proportion of CD57^+^ terminal effector cells, along with a significant decrease in the naive cell population, increased granzyme B and IFN-γ production, and unresolved inflammation 6 months after infection [195]. Specifically, T cells residing in the lung of patients with COVID-19 exhibit an exhaustion phenotype characterized by Tim-3 and PD-1 expression, which in turn may impact the severity of the prognosis [80].

### 3.5. Cytopathic Effects on the Kidney

Podocytes, proximal tubule cells, mesangial cells, and the parietal epithelium of Bowman’s capsule also express ACE2 and TMRPSS2, making nephrons and Bowman’s capsule potential targets of SARS-CoV-2 infection. Interestingly, ACE2 expression in kidneys is 100-fold higher than in respiratory organs [196]. Therefore, the kidney is susceptible to cytopathic effects due to various factors, such as direct viral action, systemic and local dysregulation of the immune system, or imbalance in the homeostasis of the RAAS system [164]. Among the cytopathic effects described in renal cells, the most frequent are vacuolization, inflammation of the rough endoplasmic reticulum, collapse of capillary tufts, effacement of the foot process, cell detachment, moderate to severe acute tubular necrosis, intraluminal debris and brush border loss in proximal tubule cells, hypertrophy and hyperplasia of podocytes and parietal epithelial cells, protein absorption droplets within the glomerular epithelium, and arteriosclerosis [197]. These cytopathic effects correlate with clinical findings in patients with renal involvement due to COVID-19, such as proteinuria, hematuria, hypokalemia with increased kaliuresis, hypouricemia, hypophosphatemia, neutral aminoaciduria, diffuse erythrocyte aggregation, obstruction of the lumen of glomerular and peritubular capillaries without platelets, fibrin thrombi or fibrinoid necrosis, red cell fragments, glomerular ischemia, myoglobin or cellular debris casts, rhabdomyolysis, and syncytia formation. All these clinical findings associated with renal lesions caused by SARS-CoV-2 have demonstrated a worse prognosis in these patients [198,199,200].

Although it is well known that COVID-19 is associated with an increased risk of post-acute sequelae in extrapulmonary tissues, renal sequelae in prolonged COVID are less well studied. One study showed that patients with recovered COVID-19 had an increased risk of renal sequelae in the post-acute phase [201]. In addition, renal alterations have been described in patients with acute SARS-CoV-2 infection, and inflammation and subclinical lesions may persist for several months after infection, with acute kidney injury (AKI) and repeated episodes of sepsis leading to progressive deterioration of renal function and chronic kidney disease [202]. Some other studies have shown long-term effects of COVID-19 on the kidney [203], and it is considered a multifactorial cause since this virus can directly infect renal podocytes and proximal tubular cells, resulting in acute tubular necrosis, protein loss in Bowman’s capsule, collapsing glomerulopathy, and mitochondrial involvement [204]. These pathophysiological mechanisms contribute to the increased mortality in hemodialysis patients infected with SARS-CoV-2 with a 12-month follow-up period [205]. Moreover, kidney transplant patients who survived COVID-19 developed general symptoms of post-acute COVID-19 syndrome, and their laboratory tests yielded altered results, including a shorter activated partial thromboplastin time and elevated levels of fibrinogen and D-dimer, all compatible with a procoagulant state [206], thus contributing to multiorgan failure in some of them, associated with significant morbidity. Regarding chronic progressive disease in patients with long COVID, endothelial dysfunction in this organ, as well as microangiopathy and alterations of the RAAS system, have been proposed as some of the main mechanisms underlying the development of this condition [164].

### 3.6. Cytopathic Effects on the Digestive System

As for the intestinal epithelium, multiple studies have shown that it can also be infected by SARS-CoV-2 since they also express the ACE2 receptor. This replication triggers the production of type III interferon by these intestinal epithelial cells, and it has been shown that this cytokine plays a key role in the control of virus replication [207]. In addition, during infection of intestinal cells, there is an altered cytokine profile characterized by upregulation of CCR1, CCR8, IL-16, IL-3, and CXCL10 (IP10), while CCR2, CCR5, and IL-5 can be downregulated [200].

In addition, syncytium formation in intestinal cells due to SARS-CoV-2 infection has also been reported, which could be mediated by TMPRSS2 and TMPRSS4. Previous studies reported that SARS-CoV and MERS-CoV could mediate this spike-mediated membrane fusion, which could be triggered by TMPRSS4, although to a lesser extent than TMPRSS2. Thus, syncytium formation in enterocytes may result in a cytopathic state that could lead to a disruption of the integrity of the intestinal epithelium [208]. Infection with SARS-like viruses can cause damage to the gastrointestinal mucosa and detachment of cell monolayers [209].

It is also important to note that the virus can alter the intestinal microbiome, thus allowing the enrichment of opportunistic pathogens. This dysbiosis can lead to alterations in the immune response, digestive tract motility, and intestinal permeability, all of which could cause the establishment of a long-term inflammatory state [24,210]. In addition, when enterocytes are infected by SARS-CoV-2, intestinal absorption through these cells is impaired, which also increases the permeability of the gastrointestinal wall. This infection also causes impaired function of mature enterocytes and overexpression of several enzymes in unusual regions, which cause damage to neighboring cells [202,210]. Similarly, liver cells can also be directly infected by the virus, which in turn leads to increased liver enzyme release, hepatocytolysis, inflammation, and, in some cases, hepatomegaly [210] (Table 2).

All of the above result in clinical manifestations associated with enteric symptoms characterized by diarrhea, loss of appetite, acid reflux, nausea, abdominal distension, flushing, vomiting, liver damage and increased liver enzymes, cholangiopathy, abdominal pain, and, in some cases, rectal bleeding [210]. In fact, patients with long-term symptoms were also reported to suffer from those symptoms during the course of the acute infection [211]. However, the long-term sequels affecting the digestive system are not present in every long COVID patient [210], although prolonged viral fecal shedding is a common phenomenon among COVID-19 patients, with viral RNA detectable in feces for several days after the end of respiratory symptoms [24]. Interestingly, a recent study reported the presence of viral proteins in the upper and lower gastrointestinal tract several months after infection in some patients, which reinforces the hypothesis of the role of a sustained immune response in the gastrointestinal system in the development of long-term symptoms [212].

## 4. Concluding Remarks

This manuscript provides a summary of the main cytopathic effects given by SARS-CoV-2 and how some of them, together with other indirect events occurring during viral clearance, participate in the development of the long-term sequelae that characterize long COVID disease. The main symptoms of this disease can be classified into two different categories: those affecting the respiratory system and those caused by damage to extrapulmonary tissues [210]. However, the pathophysiology of long COVID remains unclear, as the spectrum of symptoms varies from patient to patient. The heterogeneity observed at the clinical level could be explained by the fact that, in the absence of a consensus on the definition of the disease, patients with different severities and symptoms are often included in the same group [164]. However, previous studies have described predisposing factors for developing long-lasting symptomatology. For example, experiencing more than five symptoms during the first few days of infection correlated with a tendency to develop long COVID. Increasing age and/or body mass index, diabetes, hypertension, poor mental health, asthmatic cardiovascular disease, and female gender have also been described as other risk factors for such disease [210,213,214,215]. Thus, it is presumable that the phenotype of each patient, given by a combination of its genotype and some environmental conditions, is crucial in determining the predisposition of each individual to show long COVID symptoms. However, the SARS-CoV-2 genome also plays an important role in the development of the disease. On the one hand, certain viral genes often undergo mutations that affect the transmissibility of the virus [216]. On the other hand, the probability of developing long COVID varies according to the viral strain. For example, the incidence is higher for the Delta strain than for the Omicron strain [217]. Therefore, given that the viral genome presents a high rate of mutations, as do its proteins, it is presumable that the interaction of mutated viral proteins with that of the host cell may be altered, consequently modifying the interactome profile and, in turn, varying the cytopathic effects. As for the host genome, specific mutations and the existence of certain alleles could also predict the evolution of the disease. For example, certain variants of the IL-6 receptor have been reported to improve the outcome of COVID-19 in some patients, a phenomenon that also occurs in other proteins, such as the ACE2 receptor itself. Conversely, variants of proteins such as TLR7 could lead to severe cases of COVID-19 [218]. 

In addition, it has become clear that several organs may show long-term alterations. We review the cytopathic effects that occur in the immune, digestive, nervous, respiratory, excretory, and circulatory systems, although some other cell types and tissues such as thyroid [219], pancreatic β cells [220], and skeletal muscles [221] are targets of direct infection and contribute to the symptomatology of long COVID. However, based on the symptoms described above, the respiratory and nervous systems appear to be the two main systems affected in terms of persistent sequelae. Regarding the latter, and in accordance with the studies that reported direct infection in the CNS, it is presumable that long-term neuropathologies resulting from SARS-CoV-2 could be due to direct cytopathic effects on the nervous system and/or other indirect consequences on it, and depending on the region affected, the symptomatology will vary. Thus, the ability of individual cells to recover from SARS-CoV-2-derived cytopathy could determine the progression of the symptom in question, although further studies are needed to determine whether regenerative activity in infected regions is associated with recovery from long-term sequelae. On the other hand, several studies have hypothesized that hidden virus particles can remain in certain organs [164], which could also contribute to the maintenance of certain symptoms.

Therefore, what makes some patients more prone than others to suffer long-term sequelae? This is a question that continues to be asked by the clinic, although multiple studies are underway to search for possible causes. However, it has become clear that there are multiple factors that cause the disease, making it difficult to define. We summarize some of the evidence that SARS-CoV-2-derived cytopathy in some tissues is responsible for some of the symptoms of long COVID disease, which, in turn, is a direct consequence of the interaction of the viral and host proteomes. It is worth mentioning that not all cytopathic events occur at the same time in all individuals, as symptoms vary between patients. We believe that poor viral clearance may facilitate the virus to reach some of its extrapulmonary targets, where it can cause higher levels of cytopathy, and genetic and environmental conditions could dictate the fate of long-term symptomatology.

## Figures and Tables

**Figure 1 ijms-24-08290-f001:**
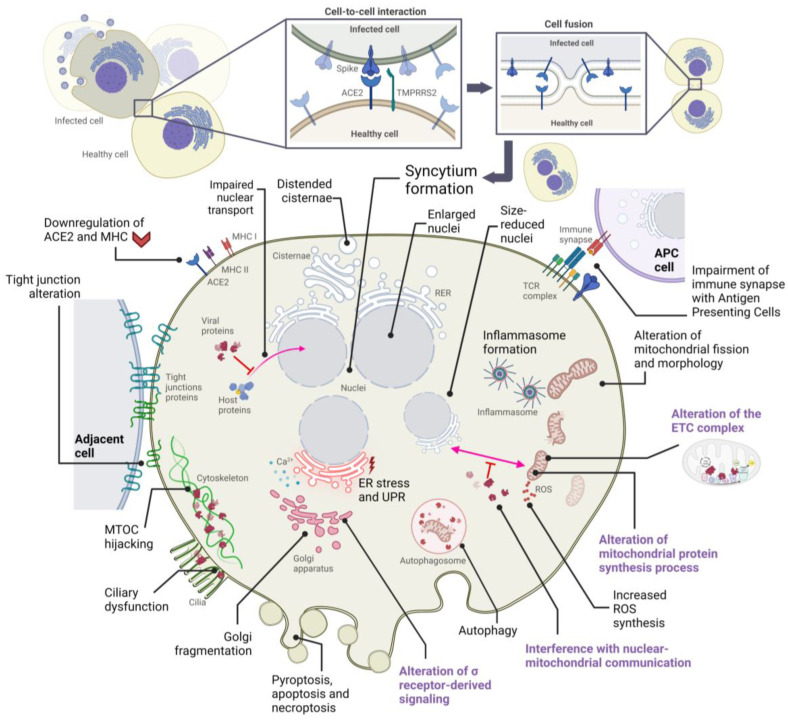
Summary of some of the cytopathic effects that occur following SARS-CoV-2 infection in specific cell types. Cytopathic effects predicted from the virus–host interactome are highlighted in purple and bold. Note that not all occur simultaneously in the infected cell, as these alterations depend on the cell phenotype.

**Figure 2 ijms-24-08290-f002:**
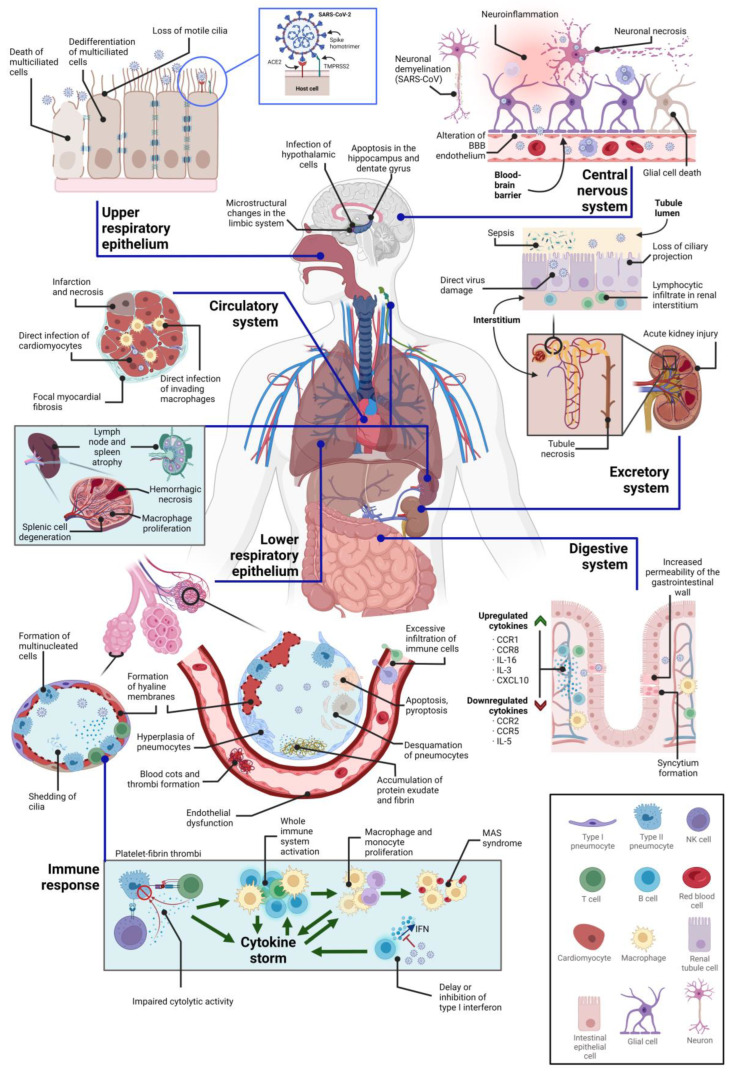
Compilation of some of the cytopathic effects of SARS-CoV-2 in various tissues, as well as some of its indirect consequences. Note that not all of these phenomena occur simultaneously in all patients.

**Table 1 ijms-24-08290-t001:** Summary of some of the most relevant cytopathic effects on each organelle that have been empirically described in SARS-CoV-2 infected cells or in cell lines transduced with viral proteins, as well as their corresponding molecular interactions described.

Organelle	Cytopathic Manifestation(s)	Affected Host Protein(s)	Responsible SARS-CoV-2 Protein(s)	References
Mitochondria	Alteration of IFN-I responses	TOM70	ORF9b protein	[39]
	Dysregulating host antioxidant defense	SIRT1	Nsp14 protein	[33,41]
Endoplasmic reticulum	ER stress response, inhibition of IFN-β	IRE1	ORF8, S, E, M proteins	[45,74]
Golgi Apparatus	Golgi fragmentation	GRASP55, TGN46	S, M, E, nsp15, ORF3a proteins	[66,68]
Cytoskeleton	Cytoskeleton remodeling	Proteins of the MTOC	S protein	[66,89]
	Ciliary dysfunction	CUL2 complex	ORF10 protein	[101]
Cell membrane	Inhibition of immune synapse	T cell receptor	S protein	[106]
Nucleus	Inhibition of transcription factor entry into the nucleus	Transcription factors (such as STAT)	ORF6, ORF3b proteins	[116]
	Inhibition of the transcription of IFN-stimulated genes	STAT, IRF3	ORF6, ORF3b, N, nsp12 proteins	[114,116]
	Inhibition of host mRNA export from the nucleus	NXF1	nsp1 protein	[119]

**Table 2 ijms-24-08290-t002:** Summary of some of the most relevant cytopathic effects affecting tissues and organs in individuals infected by SARS-CoV-2, as well as their corresponding manifestations. It should be noted that not all these alterations occur at the same time in all patients since they depend on several factors, such as the severity of the infection, age, and health status of the individual, among others.

Organelle	Affected Cell Subset(s) or Regions	Cytopathy Manifestation(s)	References
Central nervous system	Cortex, hippocampus, and hypothalamus	Neuronal apoptosis through Caspase-3	[128]
	Ventral brain blood vessels	Structural alterations	[128,129]
	Microglia	Morphological alterations	[128]
	Limbic system	Microstructural changes and volume loss	[130]
Respiratory system	Airway epithelium (ciliated cells, secretory cells, and type II alveolar cells)	Formation of viral plaques, cilia shedding and internalization, detachment of infected cells, cell death, and epithelial barrier damage	[114,131,132,133]
	Multiciliated cells of the upper respiratory epithelium	Dedifferentiation through FOXJ1 downregulation	[134,135]
	Lower respiratory tract	Hyaline membrane or platelet-fibrin thrombus formation, accumulation of protein exudates, pneumocytic desquamation, diffuse alveolar damage, atypical pneumocytic hyperplasia, and syncytia	[122,131,136]
	Type II pneumocytes	Morphological alterations of mitochondria, accumulation of lipid and protein droplets, distended ER cisternae	[122,136]
Circulatory system	Endothelium	Loss of endothelial integrity, cell detachment, endothelial dysfunction, alteration of the RAAS system, increased endothelial permeability, hypercoagulation, hypofibrinolysis, and prothrombotic states, increased mitochondrial ROS production, downregulation of ACE2 and eNOS	[24,137,138,139,140]
	Inhibition of host mRNA export from the nucleus	nsp1 protein	[120]
	Cardiac tissue (especially cardiomyocytes)	Myocardial hypertrophy, cytokine production, acute myocardial infarction, interstitial edema, contractile deficits and sarcomere disassembly, necrosis, and cell death, lymphocytic inflammation, small vessel coronary artery disease	[141,142]
Immune system	T cells	Severe lymphopenia, cytokine release syndrome, decreased IFN-γ-producing CD4^+^ T cell population, T cell depletion, alterations in T_reg_ cells, T cell exhaustion	[81,143,144,145]
	Lymph nodes and spleen	Necrosis, tissue degeneration, macrophage proliferation, atrophy	[143]
Kidney	Renal cells (in general)	Cell detachment, tubular necrosis, inflammation of the ER, alterations of parietal epithelial cells, arteriosclerosis, protein absorption droplets, syncytia formation	[146]
	Podocytes and proximal tubular cells	Hyperplasia, hypertrophy, tubular necrosis, loss of proteins in Bowman’s capsule, alteration of mitochondria, collapsing glomerulopathy	[146,147]
Digestive system	Intestinal epithelium	Disruption of the epithelium integrity, syncytia formation, digestive tract motility and intestinal permeability, alteration of the intestinal microbiome and of the immune system, impaired function of mature enterocytes, overexpression of enzymes	[24,177,185]
	Liver	Inflammation, hepatomegaly, hepatocytolysis, enzyme release	[148]

## Data Availability

This is a review article and no new data were created.

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
