# Peer review of "From Cell to Symptoms: The Role of SARS-CoV-2 Cytopathic Effects in the Pathogenesis of COVID-19 and Long COVID"

_ijms, 2023, doi:10.3390/ijms24098290_

Round 1

Reviewer 1 Report

Pablo Gonzalez-GarciaSevere et.al reviewed the syndrome of Long-COVID. The authors reviewed this syndrome from two points, one from the cellular level and the other from the organ level. From the cellular view, the author  provide a comprehensive review of the cytopathic effects of SARS-CoV-2 on various cell types, and they described the molecular mechanisms underlying virus-host interactions, including alterations in protein expression, intracellular signaling pathways, and immune responses. From the tissue view, the author highlights the potential impact of COVID on immune dysregulation, neuropsychiatric disorders, and organ damage. Finally, this manuscript explains the structure and some deficiencies in their manuscript, and provide some views on the management of Long-COVID. All in all, I think it is a good review and worth publishing. 

Before I publish, I have one suggestion. In my opinion, the order in cells and tissues is very important and should be written according to the degree of importance. The order in this paper is messy. I suggest readjusting it.

Reviewer 2 Report

This is an extremely important article. You have put a great amount of work into the literature search, the presentation of the research to date and in Figures 1 and 2 you have endeavoured to clarify the multiple pathological mechanisms. 

My comments are directed to making your article even better, more widely read and perhaps a classic article.

I counted the words "can" 120 times, "may" 91 times. "some" 72 times, "could" 56 times, "known" 15 times and "will' 5 times. This hesitancy is reflected in your conclusions: 

"However, the SARS-CoV-2 genome also plays an important role 981 in the development of the disease. On the one hand, some of its genes often undergo mu- 982 tations that affect the transmissibility of the virus [230]. On the other hand, the probability 983 of developing Long COVID varies according to the viral strain. For example, the incidence 984 is higher for the Delta strain than for the Omicron strain [231]. Therefore, given that the 985 viral genome presents a high rate of mutations, as do its proteins, it is presumable that the 986 interaction of some of these mutated viral proteins with that of the host cell may be altered, 987 consequently modifying the interactome profile and, in turn, varying the cytopathic ef- 988 fects. As for the host genome, some mutations and the existence of certain alleles can also 989 predict the evolution of the disease. For example, some variants of the IL-6 receptor have 990 been reported to improve the outcome of COVID-19 in some patients, a phenomenon that 991 also occurs in other proteins such as the ACE2 receptor itself. Conversely, variants of pro- 992 teins such as TLR7 can lead to severe cases of COVID-19 [232]."

Please re-review your literature and highlight quantitative studies which provide definite evidence of key pathological mechanisms. You may want to present this evidece in a table and also redraw your attractive figures to include key mechanisms proven to be important.

Round 2

Reviewer 2 Report

Thanks to the authors for their improvements in response to the reviewers' suggestions for the manuscript and for the table.